# Coming-of-Age Characterization of Soil Viruses: A User's Guide to Virus Isolation, Detection within Metagenomes, and Viromics

**Gareth Trubl [1],\*** , **Paul Hyman [2]** , **Simon Roux [3]** and **Stephen T. Abedon [4],\***

1   Physical and Life Sciences Directorate, Lawrence Livermore National Laboratory, Livermore, CA 94550, USA
2   Department of Biology/Toxicology, Ashland University, Ashland, OH 44805, USA; phyman@ashland.edu
3   DOE Joint Genome Institute, Lawrence Berkeley National Laboratory, Berkeley, CA 94720, USA;
    sroux@lbl.gov
4   Department of Microbiology, The Ohio State University, Mansfield, OH 44906, USA
\*   Correspondence: trubl1@llnl.gov (G.T.); abedon.1@osu.edu (S.T.A.)

**Abstract:** The study of soil viruses, though not new, has languished relative to the study of marine viruses. This is particularly due to challenges associated with separating virions from harboring soils. Generally, three approaches to analyzing soil viruses have been employed: (1) Isolation, to characterize virus genotypes and phenotypes, the primary method used prior to the start of the 21st century. (2) Metagenomics, which has revealed a vast diversity of viruses while also allowing insights into viral community ecology, although with limitations due to DNA from cellular organisms obscuring viral DNA. (3) Viromics (targeted metagenomics of virus-like-particles), which has provided a more focused development of 'virus-sequence-to-ecology' pipelines, a result of separation of presumptive virions from cellular organisms prior to DNA extraction. This separation permits greater sequencing emphasis on virus DNA and thereby more targeted molecular and ecological characterization of viruses. Employing viromics to characterize soil systems presents new challenges, however. Ones that only recently are being addressed. Here we provide a guide to implementing these three approaches to studying environmental viruses, highlighting benefits, difficulties, and potential contamination, all toward fostering greater focus on viruses in the study of soil ecology.

**Keywords:** bioinformatics; eDNA; gene transfer agent; isolate; metagenome; plasmid; ultrasmall microbes; virology; virome; virus-like particles

## 1. Introduction

Viruses are acellular infectious agents that can exist extracellularly as nucleic-acid genomes encapsidated within proteinaceous structures. These infectious agents can be found in environments wherever cellular organisms are present, particularly cellular microorganisms (microbes), such as bacteria, archaea, fungi, and protists [1]. There are an estimated $10^{31}$ viruses on Earth with a majority of them infecting microbes, making viruses significant drivers of evolution and essential for life on Earth [2,3]. Soils, as we emphasize here, are among the most virus-rich environments [4–7].

Soils represent the primary terrestrial habitat of microbes, but research on soil viruses has lagged behind research of viruses in aquatic environments. Our understanding of the soil virosphere consequently consists mostly, at best, of a 'black box', particularly in terms of the contributions of viruses to soil ecology. This underdevelopment of soil virus ecology stems mainly from complications in simply gaining access to soil viruses and their nucleic acid.

Though discouraging many from even attempting to analyze the soil virosphere, still there are multiple compelling reasons to do so (Section 2). Here, we present a "user's guide" to assessing soil

viruses, one aimed at better illuminating the soil virus "black box" by describing three prominent approaches, their pros and cons, and providing suggestions on how to reduce methodological difficulties (Sections 3 and 4). Our objective is to help researchers to better recognize what viruses may be present in a given soil sample, and to do so particularly toward assessment of the ecological impacts of viruses on different soil ecosystems.

*Virus-Like Particles (VLPs), Viruses, Microbes, and Other Terms*

Virions consist of infectious, encapsidated nucleic acid. As is customary in the field of virus ecology, however, the term 'virus-like particle' (VLP) is often used instead of 'virion'. VLPs specifically are entities of virion size that contain nucleic acid but have not otherwise been identified as viruses (described more in Sections 4.1–4.3).

In soils, viruses that infect bacteria, also known as bacteriophages (phages), are the most abundant and most studied. The term 'virus', however, is used rather than 'phage' when a virus' host has not been identified or when referring to naturally occurring viral communities, even if the most abundant viruses are phages. The term 'phage', that is, should not be used to describe the viruses of either domain Archaea or domain Eukarya [8].

To describe microbial cellular organisms, we use the term 'microbe'. Though a microbe is likely to be a bacterium due to their typically greater abundance, archaea and eukaryotes are also found among soil 'microbes'. This review thus predominantly covers approaches to determining what viruses of microbes [1,9] are present in soils, keeping in mind that most of those viruses are phages and most of those microbes are bacteria. A glossary of additional terms used in this review is found in the Appendix A.

## 2. Potential Roles and Forms of Viruses in Soils

The roles that viruses play in soils may be ecologically equivalent to the roles of viruses in oceans. There viral ecogenomics—characterizing the ecology of viruses from their genomes—and the roles of viruses in microbial biogeochemistry have been investigated to a much greater extent [10], albeit with a primary focus on double-stranded DNA (dsDNA) viruses [11]. In this section, we highlight three virus–host interactions that are potentially translatable between marine and soil environments. These include killing and lysing cells (Section 2.2), alteration of host metabolism during infection (Section 2.3), and virus-mediated horizontal gene transfer (i.e., transduction) (Section 2.4). We begin, though, with a more philosophical consideration of the utility of including viruses in considerations of soil ecology (Section 2.1), and then conclude this section with a discussion of the different ways that viruses can exist in soils (Section 2.5).

### 2.1. Importance of the Soil Virosphere

Why should we care about viruses in soils? For some, the discovery aspect alone is enough to motivate soil virus research, while others want to know to what extent viruses actually matter (e.g., in terms of their impact on soil biogeochemistry). It could be argued, however, that focus should primarily be on characterizing the metabolisms of soil organisms, whereas viruses, at least arguably, do not possess metabolisms. Rather, viruses hijack the already existing metabolisms of their hosts, especially to produce more virions, or at least to produce more virus genomes [12].

This reliance on the metabolisms of their hosts is one of the reasons why viruses technically are often not considered to be "alive" [13–15]. One result of these sometimes semantic arguments can be a focus on hosts rather than on the 'distraction' of less obviously metabolically contributing viruses. The host-centric dogma is perhaps best summed up by Fierer [16], who suggested that viruses, particularly in their role as predators of cellular organisms, represent the least important factor influencing the composition of soil microbial communities.

Contributing to this viral de-emphasis, little is known about the impacts of viruses in soils, including in terms of their roles as predators of soil microbes. Thus, the following questions should be considered:

(Section 2.2) To what extent do soil viruses impact soil microbe numbers and diversity? (Section 2.3) To what degree do viruses directly modify the metabolisms of soil microbes? (Section 2.4) How extensively do viruses horizontally transfer genes between soil microbes? Answering these questions will allow for a better understanding of the importance of the soil virosphere to soil functioning.

### 2.2. Viral Lysis of Soil Microbes

Viruses in soil ecosystems can kill, that is, serve as 'predators' of microbes. Predators, though, are traditionally considered to be organisms that kill and then consume other organisms (prey), assimilating the nutrients associated with the consumed organisms into their own bodies. Predators also tend to consume multiple prey over their lifetimes. Viruses, by contrast, exploit only a single, still metabolizing host per viral generation, and generally consume relatively little of the prey organism, though they can still assimilate 20%–30% of prey mass into new virion particles [17–20]. This is true even for viruses that are strictly lytic [21], i.e., those which can successfully replicate only in conjunction with host-cell killing. A better metaphor for lytic viruses might be that of parasitoids [22], such as larval wasps, which lethally consume their hosts alive, from within, before emerging in a mature form. Viruses nevertheless still can function across ecosystems in a predator-like manner by driving Lotka–Volterra-like predator–prey dynamics [23,24].

Is the potential of viruses to kill microbes their only aspect that matters in soil? In ocean water columns, viruses are thought to lyse about one-third of microbes each day, and in the process they collectively release about 10 billion tons of nutritious 'necromass' into the extracellular environment [25,26]. This nutrient infusion can result in an increase in numbers of existing 'osmotrophic' organisms, which in aquatic environments consist mostly of heterotrophic bacteria. The ecological result is known as the 'viral shunt' of the 'microbial loop', that is, fueling heterotrophic microbial metabolisms in part by lysing autotrophic and heterotrophic microbes [27].

While there is little empirical evidence indicating the extent to which the same microbe lysis-driven biogeochemical processes occur in soils, their occurrence in soils nonetheless seems probable. Process delays likely can exist, however, since the pore size within soils can allow nutrients to become entombed and thereby not immediately available for further microbial utilization, that is, until wetting of soils causes desorption [28,29]. Thus, we can hypothesize that viruses are important and perhaps even the main drivers of nutrient entombment within soils.

Like nutrients, a substantial fraction of virions in soils are thought to also be found adsorbed to abiotic soil materials [30]. Nutrient desorption from soil particles upon wetting therefore might be accompanied also by virion desorption. The resulting release of virions could give rise to additional infection and lysis of soil microbes, 'pumping' even more soluble nutrients into soils, analogous to the 'biological pump' in oceans [6]. Alternatively, we can speculate that, upon soil wetting, microbial replication and motility could bring cells to virions that have failed to desorb from soil particles, rather than virions desorbing and then diffusively moving toward cells. As a result, in soils some viruses could act as sit-and-wait (ambush) predators of microbes [31], rather than as diffusive 'pursuit' predators. Upon subsequent lysis, within wetted soils, substantial numbers of freely diffusing 'pursuit' virions may then be released, along with accompanying freely diffusing, lytically released 'necromass'.

### 2.3. Viral Modification of Host Metabolism During Lytic Infections

When viruses infect a host cell, often they immediately redirect that cell's metabolism toward production of virion progeny. This virus-mediated alteration in host biochemistry and physiology can directly impact microbial metabolic outputs. The extent of this impact on microbial metabolism can be higher if a virus carries auxiliary metabolic genes (AMGs). AMGs can represent more efficient versions of genes already used by microbes in their cellular metabolisms, though also can be genes which provide new functions. AMGs, though, are generally thought to be host-derived genes, representing a form of what originally were described as "vegetative viral genes" [32]. Most identified AMGs have

been found to impact central carbon metabolism and photosystem II, thus providing an immediate metabolic enhancement over other non-virus-infected cells [10,33].

AMGs are widespread in oceans but seem to be rare in soils [34–36]. Glycoside hydrolase AMGs, however, were recently identified in viruses from organic-rich soils where these genes would help break down the complex organic matter present [34–36]. AMGs thus may be less rare in soils than previously thought, a notion that can be tested as soil viruses become more sampled and characterized.

## 2.4. Virus-Mediated Horizontal Gene Transfer

Every second in the oceans there are an estimated $10^{23}$ viral infections and these infections are thought to mediate approximately $10^{16}$ transduction events between cellular microbes [25,26]. Due to the substantially greater complexity of soil environments (e.g., soil spatial heterogeneity), an equivalent soil calculation is impossible to perform. In principle, though, cellular genetic material should be similarly transferable by soil viruses between microbes, likely greatly expanding soil microbe evolutionary potential [37,38]. Transduced genes thus can provide another mechanism by which viruses can impact ecosystems, one that is in addition to their ability to phenotypically modify cellular organisms (Section 2.3) before killing and lysing them (Section 2.2).

Transduction traditionally has been viewed as a form of accidental horizontal gene transfer [39]. This generally occurs due to virus DNA-handling errors that allow host 'donor' genetic material to become encapsidated in a virion. The resulting still structurally functional virions, once released, can then deliver their accidentally packaged genetic cargo to a new 'recipient' host. Transduction usually is differentiated into generalized transduction where viruses randomly encapsidate only host DNA vs. specialized transduction, where viruses encapsidate a combination of both host and viral DNA, while usually picking up only a specific portion of host DNA [40]; see also [41].

Specialized transduction [40] along with another virus-associated horizontal gene transfer mechanism known as lysogenic conversion—which is provirus-mediated modification of a cell's phenotype that occurs during latent virus infections [42]—are both mediated solely by temperate viruses, i.e., ones capable of displaying these latent infections. Lysogenic converting genes, in contrast to donor–host genes being subject to specialized transduction, are considered to be normal components of phage genomes rather than recent accidental acquisitions. These genes can have substantial impacts on ecosystems, such as by encoding bacterial toxin genes [43]. Lysogenic converting genes are also related to, and in many cases even identical to, what are known as phage morons; extra phage genes acquired from hosts that are both stable constituents of virus genomes and expressed during virus infection cycles [44–46].

## 2.5. Many Environmental Viral States

Viruses lately have been conceptualized into two complementary states: free virions (extracellular virus particles) vs. virocells (viruses infecting host cells) [47]. It has long been known, though, that viruses are able to switch back and forth between these two states [48]. From a perspective of environmental virus microbiology, we can consider additional categories of viral states (Figure 1), and specific methods used to characterize environmental viruses will influence the degree to which each state is observed. This section presents this expanded, virus environmental-state framework (Figure 1), which builds on a simpler viewpoint considering proviruses vs. productive infections vs. free virions [34,35].

Virions are part of the encapsidated environmental fraction (category 1). Free virions usually are small in size and virions generally have genomes that are resistant to enzymatic degradation. Virions also are isolatable from unencapsidated materials and rich in viral nucleic acid.

Virocells include latent viral infections (category 2). These can either consist of host genome-integrated proviruses or plasmid proviruses. Integrated proviruses are linked to host-cell genes. Plasmid proviruses are somewhat less coupled with host genes though may be found in many copies both within individual cells and within environments. For both, the viral DNA is unencapsidated.

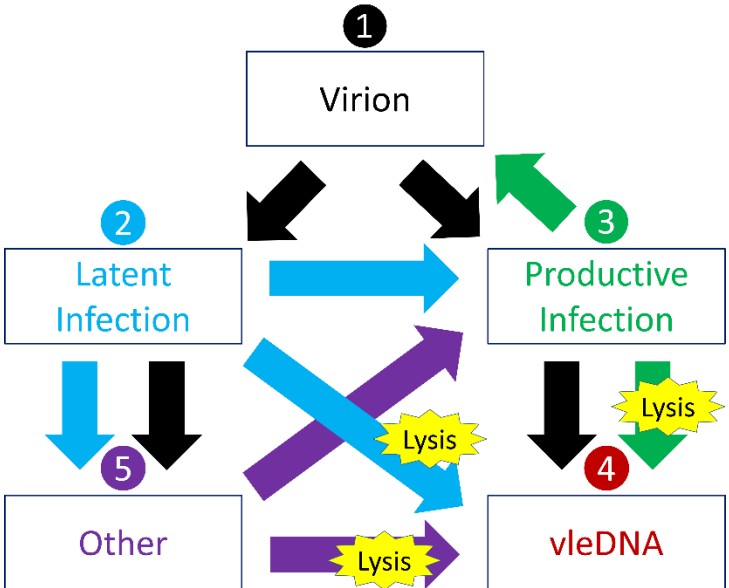

**Figure 1.** Different categories of sources of environmental viral genomic DNA. Free virions (**1**) initiate latent infections via cell adsorption (**2**) or instead initiate productive infections (**3**), where the latter can be differentiated into lytic vs. chronically releasing virions (not distinguished in the figure). Virus-like eDNA (vleDNA) is a form of extracellular, unencapsidated DNA (**4**). Viral infections also can take on various forms described here as 'Other' (**5**).

Virocells also include productive infections (category 3). Like plasmid proviruses, viral genomes undergoing productive infections usually are not physically coupled to host DNA. Unlike plasmid proviruses, productive infections in the near term are highly metabolically active, will typically generate relatively large numbers of newly replicated viral genomes, and also will generate new virions. As a result, category 3 will contain both numerous copies of a given viral genome and encapsidated nucleic acid. For lytic phages the latter will be found within particles (cells) that are much larger than individual virions.

'Virus-like eDNA' (vleDNA) (category 4) is extracellular, unencapsidated environmental DNA that has been derived in various ways from virus genomes [49]. This viral nucleic acid often is degradable using DNase and will not be physically linked to host-cell genes unless the vleDNA was derived from integrated proviruses. See Section 4.2 for further discussion of vleDNA.

We suggest an additional, catchall category of virus states that we describe simply as 'Other' (category 5). 'Other' contains viral genomes that are unencapsidated (contrasting category 1), not physically linked to host genes nor necessarily found in many copies either within cells or across environments (contrasting category 2). They are also not found in numerous copies within cells (contrasting category 3) and not derived from the extracellular environment (contrasting category 4). Examples include restricted virus genomes [50], virus infections that are unsuccessful for other reasons [51,52], viral genomes that are in a stalled pre-replicative state (i.e., 'pseudolysogenic') [53–55], or viral DNA that is contained within extracellular vesicles [56,57]. In addition, for some virus-like mobile genetic elements of fungi, no encapsidated extracellular states are even known [58,59].

Individual approaches to virus community characterization will tend to result in underassessments of virus presence or impact within environments as (i) not every viral state will be efficiently represented when using a single technique, (ii) not all detected virus-like nucleic acid will be from environmentally propagating viruses, and (iii) not all virions are easily propagated in vitro. Categories 1, 2, 3, and to some degree even 5 can, however, consist of propagatable virus nucleic acid and thereby may in principle be isolated as functional virions in the laboratory (Section 3.1). All five categories can be captured in metagenomes (Section 3.2). Only categories 1, 3, and to a smaller degree also 5 can contribute to viromes (Section 3.3). Thus, metagenomes and viromes will not consist solely of propagating viral

nucleic acid, but depending on variations in processing, can permit eDNA to be captured, and not all encapsidated DNA is necessarily of viral origin (Section 4). In contrast, not all virions are easily propagated in vitro, so viromes and metagenomes will tend to capture a greater diversity of potentially propagatable viral nucleic acid than virus isolation can alone.

## 3. Three Ways to Characterize Soil Viruses

This section describes three different methods used to characterize soil viruses. We specifically consider the pros and cons associated with each approach and how different approaches can complement each other (Figure 2). This is to provide guidance especially to researchers with less expertise on soil viruses. These methods consist of virus isolation and subsequent laboratory propagation (Section 3.1), soil metagenomics (Section 3.2), and the generation and analysis of encapsidated subsets of metagenomes known as viromes, recently dubbed as viromics (Section 3.3).

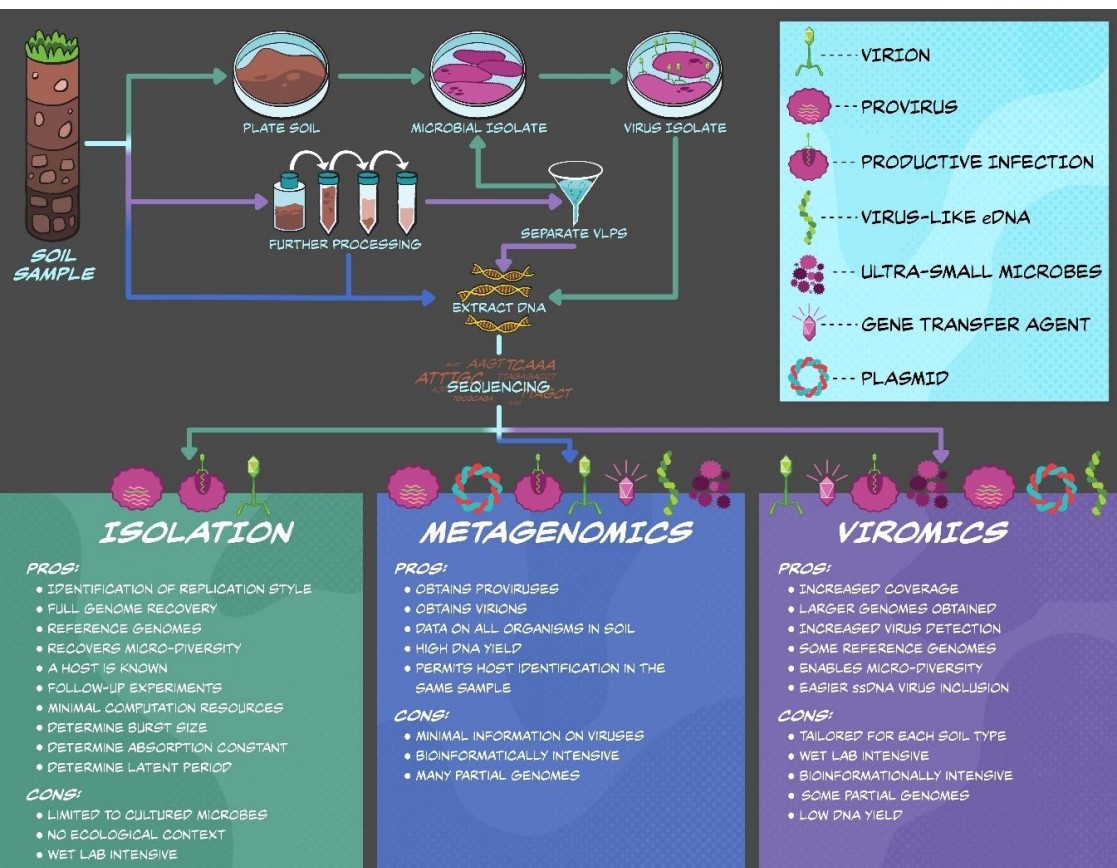

**Figure 2.** Overview of three common methods of virus characterization. Major methodological steps using virus-isolation (green), metagenomic (blue), and viromic (purple) approaches are shown. Possible contaminants, which are entities that may be described as viruses but which are not actually viruses, are denoted with icons for each approach (as summarized in Section 4). The icons are listed from left to right in order of potential prevalence in each method, although the order will depend on the sample and how it is treated. Finally, the pros and cons are listed for each approach. The pros for viromes are as relative to metagenomes.

### 3.1. Virus Isolation

Until the advent of metagenomics (Section 3.2 but see also Section 3.3), the characterization of environmental viruses first required obtaining a pure virus culture. Though essentially as old as virology itself, at least as a laboratory science, the isolation of viruses remains an important technique that offers a unique lens into understanding host–virus dynamics and can be essential toward fully

characterizing a virus' genotype and phenotype. The functions of most viral genes are unknown, and this is especially so for soil viruses [4]. The primary means of determining the function of a viral gene is by mutationally knocking out that gene and then examining the outcome of virus infection and replication [60]. Further, virus isolation enables measurement of infection metrics as burst size and latent period, and knowledge of those parameters is crucial to understanding the potential of a virus to impact an ecosystem.

### 3.1.1. Techniques for Isolating Viruses

Methods for isolating viruses from soil and other sources—especially bacteriophages—can be found in previous publications [61–64]. These include (i) direct isolation, (ii) isolation from soil wash, (iii) isolation following enrichment culture, (iv) isolation following virion concentration, and (v) isolation following induction of proviruses. For all of these approaches, after an appropriate incubation period microbial cells are removed by filtration or centrifugation (or both) and the now clarified fluid is tested for the presence of a virus. Testing typically is by plating to look for host killing as plaques, using a previously isolated indicator host [65,66]. A wealth of information and resources on culturing and characterizing virus isolates is available in the literature [67–70].

Direct plating, enrichment, and concentration: Virus isolation directly from soils or soil washes can involve simply plating using a previously isolated host strain as an indicator [71,72]. More commonly, especially when viruses are less abundant, a soil sample or wash may be incubated with a broth-cultured microbial isolate, a procedure described as enrichment culturing [61]. Even though enrichment is common practice, multiple attempts at enrichment still may be needed to obtain a single virus isolate. For example, isolation of *Mycobacterium* phages, one of the most well-cultured category of viruses, can involve 30 parallel enrichment cultures to yield one phage isolate [70,73]. This need for enrichment repetition is likely due to a combination of a high diversity of host species and strains within soils along with most viruses being somewhat host-species or even host-strain specific [51,64]. Polyvalent viruses, however, also exist, meaning that they are able to infect hosts from more than one host genus [74–76].

One can also first concentrate viruses after resuspending them from soil. This is then followed by filtration or precipitation, and only then subjecting the resulting concentrate to enrichment culturing [62]. That is, initiating enrichment cultures with potentially more virus particles from soil samples so that the number of enrichment cultures required per successful virus isolation is smaller. The initial virus resuspension step from soils is discussed more fully in Section 3.3.

Latently infecting viruses: A different approach to isolating viruses involves starting with proviruses infecting microbes isolated from soils [38,77]. In soils, approximately 30% of bacteria can harbor one or more prophages that could be induced to produce virions. The number of proviruses present in soils in fact may be even higher than that as not all proviruses are inducible using the typically employed mitomycin C [78,79]. Also, it is important to recognize that temperate phages, upon initial virion adsorption, often infect lytically rather than lysogenically [80], meaning that the same virus from the same environment could potentially be isolated using different isolation methods both as a provirus and as a virion. This ability of a phage to infect other than lysogenically commonly is described as their lytic potential, but it seems to be modifiable in response to how many hosts are present that virions can infect [81–85]. Further information on different proposed methods that bacteriophages use to regulate lysis-lysogeny conversion can be found in the literature (see [84,86–92]).

Culturing limitations: A major constraint on culturing many viruses is first growing the virus' host in pure culture, as most microbes have not yet been cultured [93]. This limitation in host availability reduces the types of viruses that can be isolated and thus studied in pure culture. Even for hosts that can be grown in culture, not all can be grown to confluence on an agar plate, i.e., so as to support the growth of virus plaques, and for some viruses, even if their hosts will readily form lawns on agar surfaces, still will not form plaques under standard culture conditions [94,95].

For those viruses that grow poorly as plaques, other approaches may be used, at least for detection, including culture clearing (culture lysis in broth) [96,97] or routine test dilution (meaning culture

clearing on a plate as near confluent lysis) [98]. Culture clearing in particular can be performed in multi-well plates in an automated system for high-throughput monitoring [99]. Finally, the original isolation host, especially as it typically will not have been isolated from the same sample as the virus isolate, is not always a primary host of a virus but instead may represent a sub-optimal host, leading to inaccurate estimation of growth parameters [100,101]. These various limitations on growing virus hosts in the laboratory make the isolation, propagation, and also ecological characterization of viral isolates in the laboratory challenging.

### 3.1.2. Well-Developed Soil-Virus Systems

A few soil virus–host systems have been particularly well developed, especially for phages and bacterial hosts. Buckling and colleagues, for example, used bacteriophage SBW25Φ2 and its soil-living host, *Pseudomonas fluorescens* SBW25, to study antagonistic coevolution between the host and phage [102] and the role of phages in host diversification [103,104]. Poisot and colleagues also used *P. fluorescens* SBW25 to isolate a variety of phages from soil washes [105]. They then looked at the range of hosts these bacteriophages could infect using bacteria co-isolated from the same soil washes, so as to examine the role of nutrient resources on the specialization of the phages. From these data, they concluded that soils which are more nutrient limited could contain phages with greater host specialization (narrower host range) than soils which, artificially, have been made more nutrient rich. Vos and colleagues [106] compared phage adaptation to specific hosts using bacteriophages and hosts that were isolated from the same soil samples. They found better adaptation of phages to local hosts—as indicated by infection rates of randomly isolated bacterial hosts—than to hosts isolated a greater distance away.

Among the most well-developed soil-virus systems are those infecting the bacterial genus, *Mycobacterium*. Mycobacteria include disease-causing along with harmless bacteria commonly found in soils. This makes mycobacteria medically relevant (e.g., *Mycobacterium tuberculosis*) as it consists of hosts for phage isolation that can be used with few biocontainment precautions (especially *Mycobacterium smegmatis*). The Science Education Alliance-Phage Hunters Advancing Genomics and Evolutionary Science (SEA-PHAGES) [107,108] and the Phage Hunters Integrating Research and Education (PHIRE) programs [109,110] successfully integrated phage isolation into mentoring young scientists and providing large collections of phages that infect Actinobacteria (the bacterial phylum that includes genus *Mycobacterium*). As a consequence, *M. smegmatis*-infecting phages represent the largest collection in the world of well-characterized virus isolates infecting a single microbial host. Recently from these efforts a patient was successfully treated with a cocktail of three phages able to infect an antibiotic-resistant strain of *Mycobacterium abscessus*, phages that were isolated using *M. smegmatis* [111].

### 3.1.3. Isolation of RNA Fungal Viruses

The methods described in the previous section presuppose that the viruses being isolated have an extracellular phase and are generally lytic to the cell. While this is true for many bacteriophages and archaeal viruses, including those with either a DNA or RNA genome, it is less frequently true for viruses of fungi, also known as mycoviruses. Mycoviruses have been identified in all major taxa of fungi, they predominantly have dsRNA genomes (although both ssRNA and ssDNA genome types exist), and for many no encapsidated extracellular states are known [58,112]. Mycovirus genomes can be isolated or otherwise identified by extracting all of the RNA from growing fungi [113] or instead using RT-PCR to target known mycovirus DNA sequences [114,115].

### 3.2. Metagenomics

Contrasting the procedures of isolation, which often focus on just a single virus or microbe clone, metagenomics involves extracting all of the DNA from a sample. The DNA is then broken up into many small fragments and sequenced (called shotgun sequencing). The resulting sequence is analyzed en

masse to reconstruct the microbe and virus genomes present. As the process does not require culturing, and most microbes cannot be cultured (as noted above), it has greatly expanded our knowledge of microbes across many environments. As it also does not rely on PCR-based detection of a universal marker gene (e.g., the microbial 16S rRNA gene which viruses do not have), it has immensely increased our knowledge of what viruses are present within environments [116–120]. With metagenomics, the composition of environmental viral communities could be described for the first time, substantially accelerating the development of environmental viral ecology.

Contrasting most notably with marine environments, metagenomic studies have not been as successful in analyzing soil viruses. The cause of this deficit has stemmed mainly from low viral DNA-extraction yields, leading to sub-optimal virus genome assemblies. The consequence is poor characterization and detection of meaningful ecological connections between viruses and microbes. As a result, most soil metagenomic studies have disregarded rather than emphasized the viral component [16]. With the advancement of new technologies to amplify lower inputs of DNA, and more sophisticated bioinformatics to analyze the sequencing data, metagenomics for virus ecology in soils is, however, becoming more feasible [34].

### 3.2.1. Losing Sight of Virus Genes in a 'Sea' of Sequence

In this section we consider various challenging aspects to characterizing the viruses found in soil metagenomes. The basis of these challenges is that there are billions of microbes found in a single gram of soil [117,121]. The vastness of these numbers boosts the appeal of soil metagenomics over microbial isolation as it is impossible to isolate all or even most of these organisms. At the same time, the resulting over-abundance of data generated by metagenomic analyses can blur our ability to finely resolve each individual type of organism, and this has especially been an issue for resolving virus genomes. Nonetheless, two general approaches to improve the virus genome-resolving power of metagenomic analyses exist: improved sequencing depth and improved sequence analysis. In addition, there usually will exist biases in terms of what DNA is sequenced or even analyzed.

Sequencing biases: The major benefit of metagenomics stems from the relatively minimal wet lab work needed before sequencing. This is in comparison with virus isolation and subsequent characterization (Section 3.1) or with separation of encapsidated nucleic acid before sequencing for viromic analyses (Section 3.3). Metagenomic analysis of a random sample of DNA nonetheless will result in biases stemming from: (i) differing abundances of community members, (ii) the specific manner in which samples are collected and stored, (iii) the physical and chemical methods used to extract and subsequently amplify DNA (the latter if applicable), and, as considered also in this section, (iv) what bioinformatic tools are used to reconstruct the metagenome [117,122]. Many of these biases, however, can be reduced with implementation especially of more standardized methodologies [123–126].

Sequencing depth: The enormous diversity of microbes and our inability to physically capture all of the DNA from a soil sample—the latter as resulting, for example, from inefficiencies in microbe lysis and DNA collection—make it impossible to sequence all of the DNA present. The DNA collected is also fragmented, which along with its high diversity makes it difficult to assemble sequenced fragments into complete or even near-complete microbial genomes, where the former, completely assembled genomes, are called metagenome-assembled genomes, or MAGs. In addition, less abundant microbial genomes tend to be not even nearly completely sequenced. The net result is that a metagenome once constructed will not be identical to the actual collection of nucleic acid sequences found in the original sample. In an effort to overcome these issues, the number of sequencing reads for a sample can be increased, which should allow increased recovery of MAGs with lower error (Section 3.2.2). This approach, however, can make assembly of abundant organisms harder due to sequencing errors that can mimic within-species (micro)diversity [127,128].

Similar assembly challenges exist for virus genomes, even though these are generally small (many thousands of base pairs) compared to the genomes of microbes, which are typically much larger, generally several millions of base pairs [129]. In addition, viruses often are not sufficiently

abundant in environments to make up for resulting differences in terms of total sequenceable DNA. Target theory [130] thus would predict a lower likelihood that a given sequencing read would 'hit' a given viral genome vs. a given microbial genome. For example, as a thought experiment, consider a 'metagenome' constructed from only a single sequencing read. The likelihood of that read being of a specific virus genome would be equal, all else held constant, to the total amount of DNA associated with that virus population relative to the total amount of DNA present within a sample; for example, about a trillion base pairs, such as $10^5 \times 10^7$ (virus genome size times number of genomes of a single virus type) vs. quadrillions of base pairs, such as $5 \times 10^6 \times 10^9$ (microbe genome size also times number of genomes of a specific microbe type). Even with somewhat more sequencing coverage, these larger genomes still can figuratively act as 'haystacks', obscuring virus genome 'needles' due to there being many more sequencing reads from microbes than from viruses. The result generally tends to be far less virus sequence and far fewer virus genomes generated in metagenomes than is the case for microbes.

Sequence analysis: Virus detection within metagenomes is further hampered by the often vast diversity of viruses present, which can make de novo assembly of viral contiguous sequences (contigs) challenging [131], that is, assembly without employing already sequenced viral genomes as templates. Specifically, the main and best assembly algorithms are based on overlapping stretches of sequenced nucleotides (i.e., De Bruijn graph assembly [124,131]), and overlapping stretches become rarer the lower the number of copies of specific viral DNA sequences that is originally present in a sample. Indeed, less than 2% of assembled sequences are typically of virus origin [34,132]. The result is decreased virus-sequence detection within metagenomes along with assembly of only partial virus genomes.

Metagenomes also are bioinformatically intensive to assemble and annotate, which can also interfere with virus identification and assembly. In particular, adding more virus detection-and-characterization bioinformatic steps can be unrealistic during metagenome analyses. Furthermore, in attempting to hunt for viruses within a metagenomic 'sea', it can quickly become apparent that virus identification itself can be non-trivial and particularly so as often most predicted genes have no annotation (Section 3.2.3) and so consequently can be difficult to assign to viruses. In total, the resulting incomplete bioinformatic 'snapshot' of what viruses are present and what specifically their genomes consist of means that virus sequence derived from metagenomes will tend to less readily reveal the functionality of what viruses are present within a sequenced environment.

### 3.2.2. Vertical Coverage

The concept of sequencing coverage can be used in two ways, horizontal vs. vertical (Figure 3). Horizontal coverage, also known as coverage breadth, refers to what portion of a contig or genome has reads aligning to it at least once, and this is often used to know how complete an assembled genome is relative to a reference genome. For MAGs, which by definition lack a reference genome, researchers rely instead on the identification of universal marker genes to estimate completeness. [133].

Vertical coverage, also known as coverage depth or sequencing depth, is by contrast the average number of reads that align to a base in a contig or an assembled genome. Vertical coverage is often used as a measure of the relative abundance of microbes or viruses within environments and can be used to determine how reliable some analyses are; for example, to assess single nucleotide polymorphisms in a microbial genome you need at least 15× coverage of that base pair [134,135]. Generally speaking, the greater the vertical coverage, the better. For instance, less abundant viruses and less abundant microbes can be missed in studies with too 'shallow' vertical coverage because a sequence consensus cannot be reached, and this can impact metagenome diversity estimates.

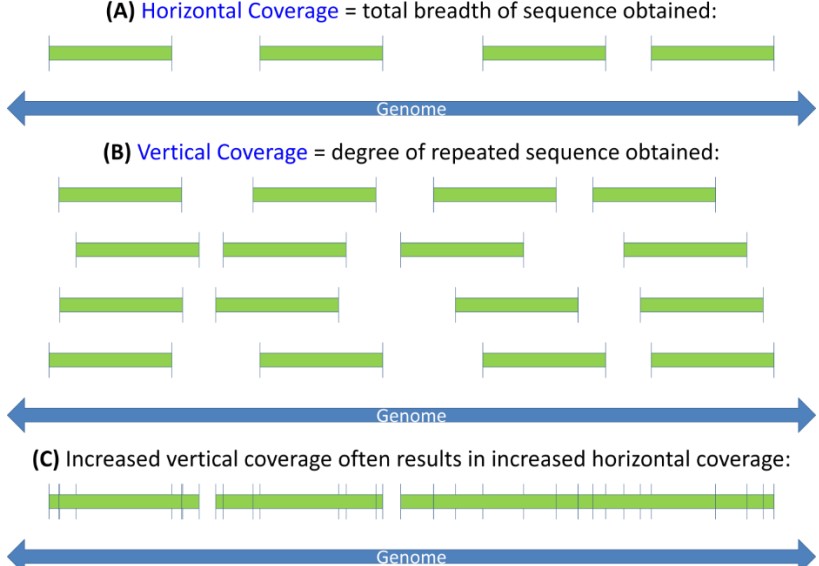

**Figure 3.** Illustration of horizontal coverage vs. vertical coverage from sequencing reads, and the impact of increased vertical coverage on horizontal coverage. (**A**) A hypothetical genome is shown as a double arrow (blue) with sequencing breadth indicated in terms of horizontal coverage of sequencing reads (green bars). (**B**) Sequencing depth by contrast is indicated in terms of vertical coverage (stacking) of overlapping sequencing reads (also of green bars). (**C**) Taking the sequencing reads providing increased vertical coverage (from B) and collapsing them into a single layer (bottom) illustrates the potential for greater horizontal coverage resulting from increased vertical coverage compared to decreased vertical coverage. Note that increased vertical coverage can also increase sequencing accuracy in terms of defining consensus sequence, though this potential increase in sequencing accuracy is not indicated in the figure.

It unfortunately is difficult to interpret the specifics of ecology from the vertical coverage of sequencing reads. Although the relative abundance of a virus in a metagenome, resulting in the potential for greater vertical sequencing coverage, suggests greater impact by those viruses on an ecosystem, higher abundance does not necessarily determine its impact in more qualitative terms. In addition, metagenomes are only a snapshot of a community and cannot provide information on community dynamics (changes over time) unless generated over a time series, an issue which is not addressed by improving the vertical coverage of only a single sample. It also simply is expensive to increase sequencing depth.

3.2.3. Drawing Information from Bulk Sequence

The primary challenge in metagenomic analysis of bulk DNA to study virus ecology is one of distinguishing viral genomic sequence from background cellular genomic sequence (Section 3.2.1). Major advancements in making these distinctions have been made including identifying viral hallmark genes (VirSorter [136]) or virus-specific motifs (VirFinder/DeepVirFinder [137,138]) to recognize likely viral contigs in metagenomes. Both types of approaches can be performed using publicly available and user-friendly programs found on CyVerse [139] or KBase [140]. These websites provide many advantages to help in the study of viruses. For example, they employ graphical user interfaces (i.e., GUIs as familiarly seen with modern computers and smart phones) rather than command line controls (the latter, e.g., as seen with the original DOS-based personal computers from the 1980s, where 'DOS' stands for disc operating system). These GUIs list hundreds of applications (apps) for processing metagenomic data along with the previous versions of those apps, and the user can sort these apps, for example, by topic or function. Additionally, optimal parameters are suggested, making analyses easier to perform, and step-by-step instructions for many of the applications are provided [141–143].

Each method has recommended conservative settings but also more-encompassing sensitive settings, determined by how likely the identified sequence represents a genuine virus (greater sensitivity, i.e., results in lower likelihood).

VirSorter, for example, uses multiple lines of evidence to place a contig into a category. Categories include 1 ("most confident"), 2 ("likely"), and 3 ("possible") for viruses not integrated into a host's genome or plasmid, with categories 4 to 6 the equivalent for integrated proviruses [123]. VirSorter relies on a database of known viral genes for category prediction. Due to this, it works especially well for marine viruses as they are better characterized genomically, but databases may be improved for virus detection in soils by the addition of new genome sequences of soil viruses, including as following virus isolation (Section 3.1). DeepVirFinder also relies on a virus reference database like VirSorter, but uses a machine learning approach in its database that enables robust detection of virus fragments ≥3 kb, with a conservative approach (likely a virus) selecting contigs with a score ≥0.9 and $p$-value <0.05 and a sensitive approach (probable virus) ≥0.7 and $p$-value <0.05. VirSorter and DeepVirFinder can also be used in parallel to optimize viral identification from metagenomic data.

A unique benefit of using metagenomic approaches is the ability to assemble viral and microbial genomes from the same data. MAGs can also be interrogated to identify proviruses. Proviruses found in high-coverage microbial genomes, integrated or not, could have increased coverage (allowing more robust analyses like micro-diversity) over viruses in other states (Section 2.5), simply due to their higher environmental prevalence within highly prevalent microbes [144]. MAGs and identified partial or complete viral genomic sequences (contigs) can be matched using several different approaches (e.g., using spacers in clustered regularly interspaced short palindromic repeats, also known as CRISPRs, and shared nucleotide identity [145]) to identify associations within the same cells and thereby possibly provide information on virus replication lifestyle [146].

Metatranscriptomic datasets can also be obtained through shotgun sequencing of RNA templates and searched for RNA viruses. While often used for assessing gene expression, genomes can be assembled from metatranscriptomes using similar pipelines as for metagenomes, and RNA virus and phage genomes can then be identified in these assemblies, including for soil samples [147]. Importantly, current pipelines including VirSorter and DeepVirFinder are not optimal for RNA virus detection due to (i) a limited number of references for environmental RNA viruses and (ii) fundamental differences in genome structure and gene content for RNA viruses; hence viral sequence mining from metatranscriptomes still requires a substantial amount of manual inspection and curation. One feature that unites all RNA viruses and can aid in their detection and characterization is their RNA-dependent RNA polymerase (RdRp). RdRps are proteins that catalyze the replication of RNA from an RNA template and are essential to RNA viruses. Analyses of RdRps consequently may provide insights into the diversity of RNA viruses and their putative hosts [148].

### 3.2.4. Outlook

Metagenomics is a powerful approach that has provided numerous insights into the characteristics of uncultured microbes and viruses, along with their possible interactions, and it continues to grow in terms of utilization. The first metagenomics papers only analyzed a small fraction of the microbial data collected and only minimal information about viruses was obtained. With the development of new computational tools and advancements in machine learning [149], however, we are now at a time where virus discovery and exploration can be performed by anyone who generates or has access to a metagenome. Notably, as newer tools become available and metagenomes become routinely generated, their sample collection and analysis needs to be more thorough [120]. This includes in terms of how samples are collected from the environment, how those samples are stored, and then how resulting sequences are documented in terms of meta data. Overall, the metagenomic approach for studying virus ecology is suitable especially for initial characterization of a soil ecosystem, for soil studies aimed at microbial diversity more generally (i.e., beyond 'just' viruses), for inferring possible

virus–host interactions, and for those just getting their 'feet wet' in terms of the study of the omics of soil virus ecology.

### 3.3. Viromics

A virome is a metagenome that consists, ideally, solely of sequence data obtained from the VLP fraction of environments (encapsidated environmental nucleic acid, also known as a VLP metagenome). Viromes are generated by separating VLPs from microbial cells, lysing those particles, and then sequencing the released nucleic acid. A virome thus can be thought of as a 'targeted metagenome', one which focuses on a specific aspect of a metagenome to better describe that fraction's specific taxonomic content and related characteristics. The first virome, published in 2002 [150], was derived from marine water and since then this approach has become the dominant method for characterizing viruses across many environments [5,10].

### 3.3.1. Utility and Drawbacks of Viromes

The main advantage viromes have over mining viral signals from less targeted metagenomes is that there is increased coverage specifically of viral genomes. That increased coverage is possible because of prior removal of the DNA of microbes and macroscopic eukaryotes. The latter, as noted (Section 3.2.1), have larger genomes that as a result are represented by a large portion of sequencing reads. The increased vertical coverage afforded by targeting the VLP fraction of biomes for sequencing therefore can yield more complete viral genomes. Consequently, greater horizontal coverage (Section 3.2.2) can increase the diversity of viruses captured, and can reveal micro-diversity within viral populations [151]. All of these benefits accumulate into complete or near-complete viral genomes that can subsequently be used as reference genomes. Reference genomes (i) are (useful for identifying new viruses from metagenomic/viromic data, (ii) can provide viral taxonomic affiliation, and (iii) can better allow for prediction of viral gene functions. Obtaining the virus fraction for targeted sequencing from soils, however, is not without challenges (Section 3.3.2).

Still, this viromics approach has many of the same drawbacks described for metagenomic studies (Section 3.2): biases associated with sample preparation; high expense (due simply to the large number of sequencing reads, though this expense is continually declining); being bioinformatically intensive [120]; and that most predicted genes have no annotation. Unlike untargeted metagenomic approaches, where DNA is extracted en masse from soil, with viromics more wet lab work is required to separate VLPs from various forms of unencapsidated soil DNA before VLP-associated DNA or nucleic acid generally can be extracted.

### 3.3.2. The Challenge of Separating Virions from Soils

The virome approach has only recently emerged as a viable option in soils, as dramatic differences between aquatic and soil environments, e.g., physical structure of soil, previously have prevented aquatic virome generation protocols from being translatable to soils. Particularly, the problem has been one of virion adsorption to soil matrix and difficulties associated with virion desorption from that matrix, at least in vitro during virome preparation. Separating VLPs from the soil matrix thus is the greatest challenge for characterizing the soil virosphere compared to viromes obtained from less complex environments, and in practice this is a time-consuming and laborious process.

More than 90% of soil viruses are estimated to be adsorbed to the soil matrix [30], and desorbing them can be tricky. There are many forces acting on viruses in soils, but the virion's isoelectric point—the environmental pH that causes a virion to have no net surface charge—is the primary factor in determining their adsorption to soil matrix [152]. It is currently impossible, however, to determine the isoelectric point of all of the virions in a soil sample. Consequently, to desorb virions, various chemical reagents with different charges and physical methods are employed [5]. Virus desorption methods in particular should be tailored to specific soil types [35,153–155]. We therefore first suggest characterizing a soil to understand its anion/cation-exchange capacity (a measure of how many ions

can be retained on soil particle surfaces [156]) and, separately, the diversity of the associated microbial community (e.g., via 16S rRNA gene surveys). The latter is because the isoelectric point of viral proteins can be strongly correlated with the isoelectric point associated with their host's proteins as has been calculated for many microbes [157].

### 3.3.3. Additional Sources of VLP Losses

VLP desorption from the soil matrix is typically followed by filtration for size fractionation. Viruses range in size from tens of nanometers (nm) to several hundred nm in diameter, making the filtration step a major point of bias, especially since most viromes are generated from viruses that have passed through a 220-nm filter [158]. This specific filter size targets phages which are typically ~50 nm in size. Because 220 nm refers to the maximum pore size, however, it is likely that even virions smaller than that cutoff may not pass through, especially larger virions [159]. In addition, if there is a lot of debris on the filter, viruses can adsorb to that rather than passing through the filter. Larger filters (≥450 nm) have also been used, but less frequently due to fear of microbe contamination (Section 4). After filtration, virions are concentrated and then these steps (i.e., chemical and physical desorption, filtration, and concentration) are repeated on the original soil sample multiple times serially to increase yields.

During these processes there is the additional issue of virions degrading or adsorbing to other surfaces after being desorbed from the soil. Virions often will adhere to every new surface encountered including those associated with the container that holds them, thereby also decreasing yields. Soils that contain a lot of organic matrix material, e.g., humic substances, are particularly difficult in that they contain an array of surface charges and matrix particles, making it hard to both desorb virions and keep them resuspended. Adsorption to organic matrix material not only changes how the virions appear, as adsorption to organic material can make it harder to identify a virion microscopically [160] (more in Section 3.3.4), but also can complicate downstream processing. For example, organic material often can bind DNA, keeping DNA in the organic layer. That DNA, as a result, is then removed from the sample during some DNA extraction methods [161,162].

### 3.3.4. Efficiency of Virus Resuspension from Soils

The proportion of viruses desorbed from a soil describes a given VLP resuspension method's virus resuspension efficiency. Different chemical and physical desorption methods can be compared by either enumerating VLPs which are endogenous to a soil sample or instead by recovering a known amount of exogenously added virus particles (the latter, also known as spike-in experiments). In virus ecology, VLP enumeration via microscopic direct counts generally is accomplished via either epifluorescence microscopy (EFM) or transmission electron microscopy (TEM). Direct quantification of VLPs from soils, however, typically is inconsistent between determinations, resulting in high variability between technical replicates and across microscopy techniques [163,164]. In this section we compare these two microscopy techniques as associated and additional approaches to determining virus resuspension efficiency.

Epifluorescence microscopy: EFM is the most widely used environmental-virus direct-count method because it is quick (sample preparation and enumerating accomplished in ~1 h, depending on the number of samples), extremely sensitive (the dyes involved strongly bind to dsDNA and RNA), and is less expensive than TEM [165]. The technique involves a combination of nucleic acid-binding fluorescent dyes and excitatory ultraviolet light that results in visualization of pinpricks of emitted light that individually correspond to VLPs. All the dyes used for EFM, however, will bind to any nucleic acid, although many preferentially bind to dsDNA. This binding promiscuity can mean that many things in a soil sample can 'light up' as VLPs during EFM, including DNA contained in extracellular vesicles [56,57] and other 'fake' viral particles [166]; see Sections 4.1–4.3 for other non-virus entities that may be part of the VLP fraction. In addition, the dye fades quickly upon exposure to the ultraviolet

light (as known as 'bleaching'), limiting the time over which a sample can be observed, although this fading can be slowed by using an antifade solution [167].

Transmission electron microscopy: TEM provides higher resolution than EFM, permitting visualization of both virus–cell interactions and virion morphology. In addition, samples can be viewed multiple times, leading to improved precision and accuracy. TEM, however, is much more expensive and time-consuming than EFM (2–3 times longer for the same sample size). It is also not available everywhere and requires considerable expertise. Furthermore, even with the high resolution afforded by TEM, it can be difficult to distinguish true viruses from non-virus particles of similar size, as will typically be found in environmental samples (i.e., non-viral or 'fake' VLPs). For an in-depth overview of TEM capabilities for viruses, see [168].

Spiking in functional virions: Both EFM and TEM can be applied to samples with either endogenously or exogenously supplied viruses. Here, we use the term, "spike-in", to describe exogenously supplied viruses. With spike-ins, virus recovery is typically measured via enumeration of plaque forming units (PFUs). In this case, the recovery of these known viruses acts as a proxy for recovery of all viruses in the soil sample [30,169]. Unfortunately, PFU enumeration is a functional rather than direct measurement, which can be misleading for efficiency determinations as it relies on virus infectivity rather than being a measurement of the absolute quantity of virus particles. In particular, viruses which become inactivated during resuspension without necessarily also losing their viral genomes will not be counted in the course of PFU enumerations, though nevertheless still will contribute to viromes.

Detecting spiked-in encapsidated nucleic acid: To focus efforts on quantification of virus particles rather than their infectivity, that is, rather than detection of PFUs, sequence-specific DNA probes that are tagged with a fluorescent dye can be designed to specifically target virions that have been spiked in, with their abundance measured via qPCR. This approach works well for quantifying known virus pathogens in the environment [170,171], but is not directly representative of native soil viruses due to these spiked-in viruses being added to an environment in which they are not endemic. These nonindigenous viruses will have different adsorption coefficients (how quickly they adsorb to surfaces) and different avidities (overall adsorption strength) for soil constituents. Thus, while quantification of these added viruses is possible, it does not necessarily translate into how well the resuspension process captures the native environmental viruses and as a result soil spiked-in approaches generally are insufficiently quantitative.

Bioinformatic approach: A different measure can be used as a metric to compare estimated efficiencies of virus resuspension after nucleic acid has been sequenced. This involves bioinformatically calculating the amount of sequencing (i.e., number of reads) of identified viruses compared to the total amount of sequencing for the sample, providing a ratio of known virus sequence to total sequenced nucleic acid [35]. A virus resuspension method can be applied to many soils or samples and the ratios determined by this approach can be compared to evaluate how well the virus resuspension method captured viruses vs. contamination. While this approach is not quantitative, since it does not measure the total number of viruses present in a soil sample, it does provide a rough measure of how much non-viral contamination is in each sample and allows comparisons of different resuspension methods and bioinformatic approaches.

### 3.3.5. Outlook

Characterizing viruses as identified from viromes has become a dominant method in the marine realm, but soil viromic efforts to date have been less rewarding. The relatively limited number of efforts to isolate viruses from soils or characterize viral genomes from soils via metagenomics or viromics has left soil-virus genetic diversity largely unknown. As a result, with each new soil virus study a majority of viruses tend be novel, which is challenging due to the difficulties in assigning functions to otherwise unknown and uncharacterized genes [172]. This usually results in insufficient recognition of virus genomes or of individual virus genes even if previously sequenced, making

untargeted metagenomic studies, in particular, less worthwhile for virome characterization. The result is minimal representation of soil viruses in current virus databases, with only ~10% of sequences in the virus RefSeq database [173] (v92) and ~3% in the Integrated Microbial Genome/Virus (IMG/VR) database [174] (v4) arguably representing soil viruses. To overcome the issue of databases mostly lacking in soil-virus sequence and the corresponding large quantity of unknown sequences in soil viromes, reference-sequence independent approaches are emerging that allow comparison of viromes that help to provide insights into spatial and temporal viral diversity [175].

In marine environments, viromics has enabled robust analyses of environmental viruses and their potential impacts on local and global ecosystems [10,176,177]. The hope is that viromics may allow equivalent characterization of virus populations in soils as has been much more readily achieved in non-soil environments. This, however, will likely be achieved only in direct association with improvements in efficiencies of virion desorption from soil matrices. No work as of now—via any of the approaches described here (isolation, metagenomics, viromics)—has characterized every type of virus that may be found in the same sample from any environment, soil or otherwise.

## 4. Metagenomic Dataset Contaminants

Above we discuss key areas for improvement of de novo assembly (Section 3.2.1), coverage (Section 3.2.2), identification of viral sequences (Section 3.2.3), and virus enrichment (Section 3.3) from metagenomics data (for more on these subjects, see [178]). In this section we consider various forms of 'contamination' of metagenomic data, that is, any environmental entities, particularly but not only VLPs, that possess a reasonable likelihood of resembling an active virus within a soil sample. Included, in order of further discussion, are: (i) non-infectious virus-like particles (niVLPs; Section 4.1), (ii) eDNA (Section 4.2), (iii) microbe contamination (Section 4.3), (iv) amplification artifacts (Section 4.4), and (v) ecologically inactive or 'banked' virions (Section 4.5).

### 4.1. Non-Infectious Virus-Like Particles (niVLPs)

Among niVLPs are otherwise intact virions which are no longer capable of successfully infecting a host, should hosts become available (contrast with phage banks; Section 4.5). Included among niVLPs are also VLPs that are not of virus origin, such as gene transfer agents (GTAs; Section 4.3.2). Non-GTA niVLPs are true virion particles which are no longer infectious due to (i) non-wholly catastrophic capsid structural damage (as still allowing inclusion in VLP direct counts but not in virus viable counts), (ii) having faulty genetic material (i.e., lethal mutations or nucleic-acid structural damage), (iii) possessing virion maturation errors (existing as incompletely formed virions) [179], (iv) having become irreversibly attached to soil components in a manner that renders them no longer cell absorbable [180], and (v) which lack genetic material due to injection into a host cell or accidental ejection into the extracellular environment. The latter, now virus capsids lacking in genetic material, would not be detected in a metagenome or virome, but could inflate VLP counts particularly as determined by TEM; dyes for TEM, such as phosphotungstic acid hematoxylin or uranyl acetate, that is, stain the virus capsid material rather than necessarily nucleic acid whereas all EFM dyes would not cause nucleic acid-lacking particles to fluoresce. See Section 3.3.4 for more on virus detection using microscopy.

### 4.2. Extracellular DNA (eDNA/relic DNA)

The vast majority of eDNA is from microbes and is ubiquitous in soils where it can play a number of ecological roles including serving as a nutrient source, as a component of biofilm matrices, or as a mediator of the horizontal gene transfer mechanism called transformation (i.e., uptake of eDNA such as by microbes). Because eDNA can persist for prolonged periods (then also known as relic DNA), it thereby can obscure our ability to characterize soil ecosystems as they exist in terms of what genomes currently are active. Though eDNA was first thought to come primarily from lysed cells, it was later determined also to be secreted by microbes [181], though it may be released from decaying virions as well; the latter a form of vleDNA (Section 2.5).

The persistence of eDNA is particularly problematic because all of the DNA within a sample is extracted to generate a metagenome. This includes from cellular organisms and viruses but also any eDNA that has persisted. It was recently shown that relic DNA in particular has the potential to inflate microbial richness estimates up to 55% depending on the soil's geochemical parameters (e.g., pH [182]). It was also recently shown in an aquatic ecosystem that eDNA accounted for about 60% of the total sequenced DNA and that a comparison of eDNA sequences to virome sequences revealed viruses that were only detected in the eDNA samples, implying that vleDNA was present [49].

Removing eDNA from Environmental Samples

In virome generation, virion purification techniques are incorporated to remove non-encapsidated DNA (e.g., DNase treatment to remove eDNA), but nevertheless non-encapsidated DNA is still detected [183,184]. One reason for this is that eDNA, including vleDNA, can be bound to inorganic or organic compounds that can prevent its degradation. Likewise, DNase requires divalent metals for activation and the presence of inorganic (e.g., copper sulfate) and organic compounds in a sample can bind divalent metals, thus partially or completely inhibiting DNA degradation [181,185]. In both cases—eDNA being protected or DNase activity being blocked—the proportion of eDNA contamination persisting past DNase purification depends on the soil composition [186–188].

New methods have been proposed to remove eDNA in the laboratory during preparation of metagenomes [182,189] or otherwise predict biases resulting from relic DNA on microbial community structure via modeling [190]. One new method to remove eDNA incorporates propidium monoazide (PMA), which is a photoreactive DNA-binding dye that can enter through pores in cell membranes, binding only to either eDNA or DNA that is found in dead cells. After a short incubation under light, bound PMA modifies DNA, preventing downstream processing, i.e., by blocking amplification and sequencing. PMA has also been used to inactivate DNA associated with damaged viruses [191], though this technique tends to be almost exclusively applied to samples containing viruses that infect humans (see [192] for a comprehensive list of studies). Presumably this technique would not work to remove all niVLPs, because, as noted, a VLP could become non-infectious due to defects in genes or nucleic acid structure rather than due to pores in capsids (Section 4.1). Nevertheless, PMA treatment is still useful, as one environmental metagenomic study, in which samples were collected from a clean-room floor, found that removal of relic DNA allowed detection of microbes and viruses that were not otherwise detected due to their low prevalence relative to that of relic DNA [155]. Once PMA treatment is performed, or any method of eDNA removal, qPCR can be used with 16S or 18S rRNA gene primers [184] to check to see if microbial DNA is still present in a virome, either because microbial relic DNA was not removed during PMA treatment or microbial DNA remained within intact ultrasmall microbial cells (Section 4.3).

### 4.3. Microbe-Derived Virome Contamination

Removal or even identification of microbial contamination in a virome is not as straightforward as it may seem. VLPs can carry rRNA genes, which is the most common way to detect microbial contamination in a virome [193] and thus genuine VLP DNA may be mistaken for more direct microbial contamination. Alternatively, microbe contamination can be incorrectly inferred if the sequences of actual viruses share similarities to known microbial sequences (e.g., AMGs or specialized transducing particles; Sections 2.3 and 2.4), which may lead to removal of sequences during bioinformatic processing and thereby loss of legitimate virus data. On the other hand, microbial DNA may represent contamination stemming from the presence of ultrasmall or dormant microbes possessing decreased cell size (Section 4.3.1), GTAs (Section 4.3.2), or even plasmids that may or may not encode virus genomes (Section 4.3.3). Though not discussed further here, note that the converse of small microbes being similar in size to typical VLPs, is large VLPs being similar in size to typical microbes [194].

### 4.3.1. Ultrasmall Microbes

Ultrasmall microbes, those that can pass through a 0.45-μm filter, and some even 0.2-μm filters [195], are widespread and are found in the Bacteria and Archaea domains. Though not the same, ultramicrocells also exist, which are microbes with reduced cellular size due to dormancy as may be induced for various reasons including starvation [196]. The small cell size of ultrasmall microbes is matched by small genomes that do not include non-essential DNA, resulting in reduced functional potential [197]. Many ultrasmall microbes actually do not have enough metabolic capability to survive in isolation, e.g., as due to some missing complete housekeeping biochemical pathways. Instead, they join with other microbes to form metabolic networks.

Part of bioinformatic virus detection (described in Section 3.2.3) is identifying motifs typically exhibited in viruses, including enrichment of uncharacterized genes or possession of short genes, things which ultrasmall microbes can also exhibit [197]. Ultrasmall microbes thus can be similar to viruses in their genomic properties, which can make them a challenging virome contaminant to remove. Ultrasmall microbes nonetheless are not likely to be present in appreciable quantities in metagenomes for the same reasons that many viruses are also not present in appreciable quantities (i.e., their smaller genomes) (Section 3.2.1). In addition, unlike viruses, ultrasmall microbes can be detected bioinformatically because of their 16S rRNA genes. Nevertheless, due to their small size, ultrasmall microbes can be mistaken for viruses during EFM-based direct counts (Section 3.3.4), thereby inflating perceived VLP numbers.

### 4.3.2. Gene Transfer Agents (GTAs)

Marine viromes have been rigorously optimized yet still can contain presumptive cellular DNA contamination comprising approximately one third of metagenomic sequencing. This genetic material presumably is of non-viral origin and otherwise is thought to consist mostly of DNA carried by GTAs [198,199]. GTAs are non-viral though nevertheless are VLPs, containing pieces of genetic material obtained from the genome of the microbe they originated from and which they can transfer to other, similar microbes. GTAs, unlike true viruses, cannot however directly create progeny GTAs [200]. GTAs nonetheless have been proposed to be atypical, genetically defective viruses, or viruses that have been otherwise repurposed by a host particularly to horizontally transfer host DNA [201]. In any case, GTAs represent a form of niVLPs. Indeed, to detect GTAs, many studies have focused on the genome sequence characteristics of what microbes are most likely to produce GTAs along with common genotypic characteristics found among identified GTAs, i.e., particularly possession of few if any known viral genes [200].

Currently, more GTAs have been identified from *Alphaproteobacteria* than any other group of cellular microbes [200]. This consequently presents a potential problem for soil viromics because these bacteria are some of the most abundant microbes found in soils [202]. For example, a recent study proposed that GTA-associated genetic material, based on sequence similarity to *Alphaproteobacteria* DNA, can represent up to 25% of assembled reads from viromes generated from peat soils [35].

### 4.3.3. Plasmids

Plasmids are extrachromosomal, semi-autonomous, either circular or linear pieces of DNA, and they are present in most microbes [203]. They regularly encode genes that are non-essential to their cellular hosts, i.e., as known as accessory genes. Most notably from a medical microbiology perspective, this accessory genetic material includes antibiotic resistance genes. Plasmids can move between microbes during conjugation (particularly bacteria connecting via sex pili, effecting DNA movement), via transformation, and by transduction [204] (for more on transduction, see Section 2.4).

Plasmids and viruses can have many similar genes, especially for DNA replication and interaction with host defenses [205,206]. Plasmid DNA sequences, unlike those of viruses generally, are also common in metagenomes and present problems for virus identification as undertaken via automated

viral detection, i.e., as due to plasmids encoding virus-like genes [136,207]. For instance, VirSorter (Section 3.2.3) detects viruses based on viral hallmark genes, which can also be picked up by microbial hosts and transferred into plasmids. Discerning between a virus- and a plasmid-encoded virus-like gene within a metagenome also can be difficult because most of the genes in question may be unknown and only genes that are known and previously associated with viral genomes may be described with any certainty as viral genes. Thus, a plasmid with virus-like genes can easily be identified as a virus. New bioinformatic tools, however, are being developed to detect plasmids in metagenomics datasets either for removal or for use in plasmid-focused investigations [208–210].

In terms of plasmid inclusion in viromes, it is important to note that plasmids are not encapsidated and are thereby mostly excluded from viromes. Plasmids also can represent a component of eDNA, but like vleDNA, plasmids should be excludable from viromes to the extent that eDNA is removed, for example by DNase treatment, prior to removing encapsidated DNA from viral capsids.

### 4.4. Amplification Artifacts

Viruses with single-stranded DNA (ssDNA) genomes are diverse, ubiquitous, and infect all domains of life including numerous microbial taxa [211]. The study of ssDNA viruses has arguably benefitted the most from metagenomics due to its greatly expanding the number of known ssDNA viruses, cataloging the hosts they infect, and highlighting their environmental roles, all paving the way for a global analysis of ssDNA viruses and their importance [212]. Even with the advent of metagenomics, however, ssDNA viruses are tricky to both detect and study because they can have segmented genomes, which can appear as separate viruses in metagenomic datasets. Investigations are further impeded because ssDNA viruses undergo rapid mutation while evidence supports widespread horizontal gene transfer [212]. The importance of ssDNA viruses in environments nevertheless may be overstated.

The paradigm in question is that ssDNA viruses are the most abundant virus type in soils. This conclusion, however, appears to have arisen partially because of the need to greatly amplify viral DNA for the sake of generating sufficient quantities for sequencing. Many whole genome amplification methods have been used to overcome the issue of low DNA yield extracted from environmental samples (e.g., random priming-mediated sequence-independent single-primer amplification [159,213]), but multiple displacement amplification (MDA) was the most widespread whole genome amplification method implemented until the 2010s. This technique uses rolling-circle amplification, which has been shown to preferentially amplify circular ssDNA, including that of plasmids, while unevenly amplifying linear genomes [214]. The result is a biased inflation of the abundance of ssDNA viruses in samples, making their actual abundance unknown and quantitative comparisons to other datasets thereby impossible. Thus, while ssDNA viruses are of interest, they are unlikely to be as prevalent as earlier reports suggested.

While it has taken some time, whole genome amplification methods are being replaced with methods that quantitatively capture ssDNA viruses and permit ecological comparisons between ssDNA and dsDNA viruses, providing a more holistic view of the soil virosphere. The first important development was to optimize the first step of traditional high-throughput DNA sequencing protocols (adapter ligation) to allow for PCR amplification and accurate sequencing of both ssDNA and dsDNA [215]. The initial aim of these modifications was to increase the accuracy of sequencing, and because most living things have dsDNA genomes (ssDNA viruses, of course, excepted), adapters were designed to aid in the sequencing of both strands of DNA from a single molecule [215,216]. A library method also was recently developed that included novel adapter attachment chemistry, which permits quantitative amplification and sequencing of ssDNA, dsDNA, and damaged DNA in parallel [217]. To test its fidelity, this method was first applied to mock viral communities [214] and since has been shown to capture both ssDNA and dsDNA viruses in many environments including soil from picogram-level input DNA [155,218]. Library preparation kits and protocols able to generate

quantitative metagenomes from nanogram DNA inputs are thus now readily available and should be primarily used, as opposed to non-quantitative amplification.

To aid in the detection of ssDNA viruses, many studies have utilized the fact that the majority of ssDNA viruses are circular and encode known marker genes, such as homologs of genes encoding the rolling-circle replication-associated protein [36,155,206,219]. To date, there has only been one study that quantitatively amplified both ssDNA and dsDNA viruses from the soil samples [155]. Using known ssDNA virus marker genes, it suggested that ssDNA viruses were a small fraction of the microbial viruses observed (~4%). To fully evaluate this 'ssDNA viruses are dominant in soils' paradigm, however, additional quantitatively amplified soil viromes are needed that evaluate the relative abundances of ssDNA to dsDNA viruses, with a careful consideration of contaminants, as contaminants can increase perceived abundances of dsDNA viruses, e.g., all known GTAs and ultrasmall microbes carry dsDNA rather than ssDNA [200].

*4.5. Ecologically Inactive Viruses*

Functionally active but nevertheless ecologically 'inactive' viruses can be described as being in a 'Bank mode', as equivalent to 'Seed banks' for plant populations. This ability vs. inability to potentially cause future infections distinguishes, respectively, banked viruses from niVLPs. The banked mode concept further proposes that only the most abundant viruses within an environment are likely actively replicating [220].

In soils, viruses in banked mode arguably exist as two different subcategories: (i) functionally active viruses that cannot reach a host for a variety of reasons including reversible adsorption to soil matrix, and (ii) functionally active viruses that infect only rare hosts. In the first case, these viruses could become ecologically active when environmental conditions change; for example, when rainfall creates channels in the soil matrix permitting movement (Section 2.2). In the latter case, these viruses are always ecologically active, because their hosts remain, filling a niche. Actually, banked viruses are still ecologically important, at least over longer time frames, because they can help maintain the diversity of viruses, but are nearly impossible to distinguish within metagenomes or viromes from viruses that are more ecologically active. They might be distinguishable instead in metatranscriptomes, time-course experiments, or in experiments where active viruses are labeled (e.g., stable isotope probing [221]).

## 5. Conclusions

The still young field of soil virus ecology deserves continued and indeed enhanced attention as soils are a central component of many of Earth's biomes, and viruses are increasingly recognized as important to ecosystem functioning. Different approaches to the study of virus ecology, however, have not been equivalently developed. This can result, in some cases, in intellectual biases where seemingly 'better' data come to dominate thinking even if it also underlies different, potentially competing, and not necessarily superior perspectives. Nonetheless, and despite disparate efforts to date, our understanding of the roles of viruses in soils remains meager on nearly all fronts, leading to the functional and ecological importance of viruses in soils to be largely overlooked.

In virus ecology, intellectual biases can perhaps be seen especially in terms of genotypic (sequence-based) characterizations vs. characterizations that are more phenotype-based. This can particularly be the case since sequence-based characterization is often easier to perform and certainly can provide far more data that are more straightforward to analyze using computers. The study of soil virus ecology nevertheless may be relatively unique in this regard in that sequence-based virus analyses, particularly viromics, can also be somewhat difficult to perform with soils, owing to the complexity of the soil environment physically, chemically, and spatially. That is, despite the growing torrent of sequence-based soil viromics data, its role in our understanding of soil virus ecology remains somewhat underdeveloped.

Here we outlined various approaches to undertaking both phenotypic and genotypic characterizations of soil viruses, including the challenges and solutions, with emphasis on improving

sequence-based characterizations. Given that soils lag behind other environments in terms of the development of viromics and virus ecology, an important near-term emphasis should be on improving omics approaches in soils and consideration of viruses in all soil microbiome studies.

**Author Contributions:** G.T. was the primary writer of the manuscript. S.R. and P.H. provided editing and made contributions to some of the writing. S.T.A. was invited by *Soil Systems* to provide the article, contributed to the writing, and otherwise served as senior author on the manuscript. All authors have read and agreed to the published version of the manuscript.

**Funding:** Portions of this work were written under the auspices of the U.S. Department of Energy by Lawrence Livermore National Laboratory under contract DE-AC52-07NA27344. The work conducted by the U.S. Department of Energy Joint Genome Institute is supported by the Office of Science of the U.S. Department of Energy under contract no. DE-AC02-05CH11231.

**Acknowledgments:** We graciously thank Noah Sokol, Rachel Hestrin, Eric Slessarev, Aram Avila Herrera, and Dinesh Adhikari for their feedback on the manuscript. We thank Ryan Goldsberry for making illustrations for the graphical abstract and Figure 2. A special thanks to Ella Sieradzki for doing a courtesy review of the manuscript.

**Conflicts of Interest:** S.T.A. generated and maintains various virus ecology-emphasizing web pages, including phage.org (the Bacteriophage Ecology Group). The other authors declare no conflicts of interest.

## Appendix A. Glossary of Terms

**Antagonistic coevolution**. Interactions between two species in which evolutionary adaptations in one species negatively affect a second, resulting in evolution of counter adaptations by the second species; for viruses this is typically seen as a coevolutionary arms race where host organisms evolve virus resistance which is then overcome by virus adaptations.

**Auxiliary metabolic gene (AMG)**. Gene encoded by a virus that was acquired from a previous host organism, and that can be expressed during virus infections to alter an infected cell's metabolic activity over that of uninfected cells.

**Bank mode.** Refers to virions that are dormant but not inactive, particularly due to current lack of access to absorbable host organisms, but with the dormant state potentially reversible once an absorbable host appear.

**Biogeochemistry**. Biological, chemical, geological, and physical processes that occur in an environment particularly involving movements of nutrients within and between ecosystems.

**Biome**. A community of organisms occupying a major habitat.

**Chronic release**. Virus infections in which virions are released without substantial disruption of host cells, for instance as via virion extrusion or virion budding across or from the host-cell envelope, in contrast to lytic infection.

**Community**. Multiple species living together in a given area.

**Confluent lysis**. The inability to delineate where one plaque ends and another begins, making a plate appear to be completely covered by interconnected plaques. This is typically the result of plating too many virus particles that are too numerous to count.

**Contiguous sequence (contig)**. Referring to nucleic acid sequences that are adjacent within the genome of a single organism; sequencing reads that can be assembled into a larger genome fragment are ones which are contiguous.

**Coverage**. Bioinformatics term that describes the extent to which an assembled genome has sequencing reads that map to it, either across the genome or to a specific region; this can be differentiated into coverage breadth (or horizontal coverage) vs. coverage depth (or vertical coverage).

**Coverage breadth**. Proportion of a genome to which sequencing reads align, that is, the fraction of a genome that has been successfully sequenced; also known as horizontal coverage.

**Coverage depth**. Number of sequencing reads that map to a specific region of the genome, that is, the degree of sequencing redundancy achieved; also known as vertical coverage.

**De novo assembly.** The assembly of a contig using an algorithm, instead of assembly using a reference genome.

**Desorption**. The release or detachment of a substance or particle from a surface.

**Ecogenomics**. Methods of determining ecological characteristics and interactions from genome-sequence information.

**Enrichment culture**. Technique toward amplifying microorganisms of specific phenotype from an environmental sample; consists, for viruses, of adding the sample that might contain a virus to media along with specific host cells to allow amplification of virus numbers.

**Environmental DNA (eDNA)**. DNA that is present in an environment outside of a biological entity, which for most microbes is extracellular.

**Epifluorescence microscopy (EFM)**. Imaging technique that uses a microscope that emits light in ultraviolet wavelengths to cause fluorescence of parts of a specimen.

**Extraction**. See viral extraction.

**Gene transfer agent (GTA)**. Virus-like particle that is not biased toward packaging the DNA responsible for producing it but rather packages all cellular DNA with roughly equivalent probability.

**Generalized transduction**. Process by which DNA is moved from one host to a different host due to a virus accidentally, randomly encapsidating host DNA without associated viral DNA; contrast with specialized transduction and gene transfer agents.

**Hallmark genes**. Genes in a viral genome that are central to virus replication and structure, and are shared by a broad variety of viruses, but are missing from cellular genomes.

**Horizontal coverage**. Synonymous to coverage breadth.

**Horizontal gene transfer**. Movement of genetic material between organisms other than in the course of either reciprocal sexual gene exchange or vertically from parent to offspring; virion-mediated horizontal gene transfer generally is called transduction.

**Induction**. As pertaining to proviruses, the transition from an established latent cycle to a productive infection, including as can be forced, e.g., as in the course of mitomycin C treatment of bacterial lysogens.

**Integration**. Process of insertion of a provirus' genome into existing host genomic DNA, within a host cell, as toward establishment of a latent infection; integrated proviruses become physically linked to host genetic material; contrast with plasmid provirus.

**Isolation**. See viral isolation.

**Latent infection**. Virus infection during which virion progeny is not produced but viral genome replication occurs.

**Library**. In the context of metagenomics, a library is a DNA template prepared for sequencing, including following amplification of DNA to adequate levels for sequencing.

**Lysogen**. Especially a bacterium harboring a prophage; that is, a bacterium hosting a lysogenic cycle.

**Lysogenic conversion.** Virus-encoded modification of a cell's phenotype that occurs during latent virus infections and is not a result of normal virus functioning. Lysogenic conversion is not directly associated with retention of the latent-infection state; lysogenic cycle repressor genes, for example, therefore are not also converting genes.

**Lysogenic cycle**. Ongoing, especially bacteriophage existence as a prophage; a bacteriophage latent infection.

**Lytic cycle**. Productive viral infection which ends with virion release via host-cell lysis.

**Maturation error**. Failure of virion components to properly assemble into an infectious virus particle.

**Metagenome.** Collection of sequences obtained from untargeted sequencing of all nucleic acids extracted from a biome sample.

**Metagenomics.** Non-culturing set of method to extract, sequence, and analyze a portion of all nucleic acid from a biome sample.

**Metagenome-assembled genome (MAG)**. Near complete to complete genomes of organisms assembled from metagenome sequence information.

**Microbial loop**. The movement of nutrients, especially carbon, from a dissolved state in an environment up through multiple microbial trophic levels, particularly movement from dissolved organic carbon to heterotrophic bacteria to protozoa.

**Micro-diversity**. Genetic diversity among individuals within a population (same species).

**Mitomycin C**. Chemical that alkylates DNA and forms cross-links, causing significant cytotoxicity to cells and resulting in SOS responses and associated induction of proviruses.

**Mobile genetic elements**. Any entity that moves nucleic acid between loci either within or between cells or organisms.

**Moron**. Gene acquired from host cells especially as are expressed by viruses during latent cycles but whose function is not necessarily directly related to virus metabolism during latent or productive infections, i.e., as representing 'more' DNA.

**Multiple displacement amplification (MDA).** An amplification technique that uses a polymerase isolated from phi29 bacteriophage to generate sufficient quantities of DNA for sequencing.

**Necromass**. Total mass associated with dead organisms in an environment or sample.

**niVLP.** Non-infectious virus-like particles.

**Non-infectious virus-like particle (niVLP)**. VLP that is incapable of infecting a host organism.

**Osmotrophic**. Referring to organisms obtaining energy and nutrients from dissolved environmental materials, e.g., with heterotrophic bacteria and fungi serving as key osmotrophic organisms in soil environments.

**Plasmid provirus**. A virus that replicates separate from host chromosomes while latently infecting.

**Predator**. Organism that kills other (prey) organisms in order to obtain nutrients from that other organism's now-dead body.

**Prophage**. Bacteriophage provirus.

**Productive infection**. Viral infection in which new progeny virions are produced and released, the latter either lytically or chronically depending on the virus.

**Provirus**. Latently infecting virus genome as present in a host cell.

**Pseudolysogeny**. Virus infection which has stalled including as due to nutrient limitations but that is capable of restarting toward either a productive or latent infection.

**Read**. Short for sequencing read, i.e., genotype information of an organism obtained via one individual nucleic acid sequencing process.

**Reference genome**. A representative example of an organism's nucleotide sequence.

**Relic DNA**. DNA that has been preserved in an environment in a non-functional form over extended time periods, e.g., more than seconds, minutes, or hours; see also, for example, niVLPs and vleDNA as well as eDNA.

**Restricted infection**. Virus infection that cannot be completely executed, thus interfering with virus propagation but not necessarily in which the virus-infected cell is inactivated/killed; for example, as mediated by bacterial restriction-modification systems.

**Resuspension**. See viral resuspension.

**Richness**. Number of different populations (different species) found in a given area; short for species richness.

**Rolling-circle amplification**. In vitro nucleic-acid replication process in which multiple copies of a circular template are generated by a polymerase using one nucleic-acid strand as template while displacing the other strand, i.e., as based on rolling-circle replication.

**Sequencing depth**. Synonymous to coverage depth.

**Shotgun Sequencing**. A method where DNA is broken up into many small fragments, which are then sequenced in parallel to obtain multiple overlapping reads to determine the original DNA sequence.

**Soil wash**. Process where a buffer solution is added to a soil sample, mixed, and the sample then centrifuged, with resulting supernatant recovered.

**Specialized transduction**. Process where DNA flanking an integrated provirus is encapsidated after an error in provirus excision with it along with virus genomic material then transferred to a new host cell; contrast with generalized transduction.

**Strictly lytic**. Virus that upon infection is inherently unable to display either latent cycles or chronic release; synonymous with obligately lytic.

**Structural damage**. Irreversible physical disruption of a virion particle capsid or appendages; contrast with genomic mutation or nucleic-acid damage.

**Targeted metagenome**. Metagenome generated with specific, biasing steps to focus on a subset of a community; viromes, for example, are targeted metagenomes.

**Temperate virus**. Virus that can perform both latent and productive replication cycles, though not both at the same time.

**Transduction**. Process of horizontal gene transfer between cells that is virus effected.

**Transmission electron microscopy (TEM)**. Technique that uses a beam of electrons rather than light to illuminate a specimen and thereby create high resolution images (micrographs); TEM can be used to visualize virus particles within environmental samples.

**Viral extraction**. A process to lyse viral capsids to release the DNA using a combination of physical and chemical methods.

**Viral isolation.** A process whereby a single virus is propagated in the laboratory in association with its host.

**Viral metagenome**. Targeted metagenome focused on viral (or VLP) nucleic acid sequences from a biome sample.

**Viral resuspension**. Process to desorb virions from soil using a combination of physical and chemical methods.

**Vertical coverage**. Synonymous with coverage depth.

**Viral shunt**. Solubilization of cellular organisms, especially microbes, via virus-induced lysis, thereby preventing or delaying energy and organic carbon movement from these organisms to higher trophic levels.

**Virion.** A complete infectious virus particle including nucleic acid, a capsid, and sometimes an envelope.

**Virome**. Synonymous with viral metagenome, i.e., a metagenome that has been biased toward sequencing of the VLP portion of biomes.

**Viromics**. Targeted metagenome method with specific steps to sequence especially viral nucleic acid from a biome.

**Virosphere**. All of the viruses found in a given area.

**Virus-like eDNA (vleDNA)**. Environmental DNA that is either from or thought to be from a virus, i.e., eDNA of probable virus origin.

**Virus-like gene**. Genetic material not necessarily explicitly from a virus source that is the best match to a known virus gene and/or which is localized with nearby virus genes to a specific strand of DNA.

**Virus-like particle (VLP)**. Particles of virus size as found in an environmental sample that potentially contains viral nucleic acid, i.e., something that probably is a virion particle but is not necessarily a virion particle; an alternative definition, from the medical virology literature and not used here, is a virus capsid that lacks viral nucleic acid.

**Virus-specific motif**. Nucleic acid sequence pattern that is indicative of a virus; see also virus-like gene, i.e., the best match to a known virus gene, or otherwise known virus nucleic acid pattern.

**VLP metagenome**. Synonymous with viral metagenome.

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
