# Peer review of "Coming-of-Age Characterization of Soil Viruses: A User’s Guide to Virus Isolation, Detection within Metagenomes, and Viromics"

_soilsystems, doi:10.3390/soilsystems4020023_

Round 1
Reviewer 1 Report
The manuscript ‘Coming-of-Age Characterization of Soil Viruses: A User’s Guide to Virus Isolation, Detection within Metagenomes, and Viromics’ provides an up to date review of recent advances in our understanding of viruses in the soil. The paper is detailed and comprehensive, and with a notable exception (see below), covers this growing field adequately. In fact, in some places, there is too much information, and some of the information is not always relevant to soil viruses, but reads more like a literature review of a thesis (see below for more details).
The writing style is a little lengthy. In many places, the sentence structure is unnecessarily complicated, or does not make sense. The authors should look out for places where words such as ‘thus’, and ‘hence’ are used for no real reason. I have tried my best to include some of these below. The manuscript would benefit from a more in-depth proofread from all authors. I found many typos, and this is always disappointing as a reviewer.
A major drawback is the omission of recent advances in metatranscriptomics, most notably, Starr et al. (2019) (https://www.pnas.org/content/116/51/25900). I suggest that a section on metatranscriptomics is included in the text, as well as in the figures. In fact, the paper omitted RNA viruses totally, and some thought should be given to how these advancements may impact the future of this field.
The term VLP is problematic. There is no agreed upon definition or VLP. The term is mostly used to refer to particles that have the morphology of a virus, typically observed by TEM. The term is also used to refer to synthetic self assembling proteins, and assembled capsids that lack nucleic acids, which is in direct conflict with the authors definition. Due to the lack of an agreed definition, perhaps the term is best avoided altogether.
The section on vertical coverage is not really specific to soil viruses, and feels a little basic. E.g. ‘The actual DNA sequence could read ACCGT, but the computer analyzing it may read ACCCT’. Do we really need this kind of discussion in this review? If you wanted to make it more interesting, you could model figure 3, using randomly generated contigs,instead of a cartoon figure. I don’t really know if all this is necessary here. The review is already quite long, so maybe some editing is in order. Similarly, do I really need to be given examples of command line controls vs GUI. Again, there is lots of detail on VirSorter, why not just cite the paper and discuss how useful it is in the context of soil viruses.
Overall, I believe with some polishing, this will be a valuable addition, and of real use to other scientists looking to enter into this growing field.
Below I list essential improvements:
Ln 18: Awkward syntax “but is still limited by an obscuring of viral signals by cellular DNA”
Ln 42: Poor syntax
Ln 69: add comma after ‘section’
Ln 88: hyphen in de-emphasis (I think, but look in a dictionary...)
89: I find the placement of the ‘(Section X.X)’ in this section distracting. These should be at the end of each sentence, not after the first 3 words.
- “Therefore might be… as well” is a bit complicated. Try deleting the ‘therefore’ and the ‘as well’.
- The use of ‘Such’ at the start of the sentence seems unnecessary. Similarly, ‘as now more or less’ is wordy, and ambiguous. Simplify this sentence.
- ‘I.e. as analogous to the…’ is also overly wordy.
- Change to: Alternatively, we can speculate that, upon soil wetting, microbial replication and motility could… etc… Maybe try and split this into two sentences to make it easier to understand.
- Grammar: ‘In soils, that is, some viruses…’. This reads horribly.
- Is ‘nevertheless’ adding anything to this sentence?
- Do you really mean appreciated. Appreciated by who?
- Find a better way of saying ‘virus-infection-helpful’
- What kind of complexity do you mean. Be specific, the oceans are also complex in their own way.
- Give more of a definition for lysogenic conversion. The bioinformaticians will not know what you mean!
- ‘that latter as found in what are known instead as transducing particles’. This does not make sense
- Use of ‘such’. Is it necessary?
- ‘Particularly as evolved phage functions’. This doesn't really make sense to me. Revise sentence structure.
- Employed. What exactly do you mean to say here? Expressed?
- Virions and virocells?
- ‘Are part of’ instead of ‘contribute’.
- Can you include a reference for vleDNA? How do you prove it was not encapsidated after is has been sequenced?
Figure 1. What about RNA viruses?
Figure 2. Bioinformationally is not a word? Please explaiun gow you determined the prevalence (and order) of the different icons. Do you have data to support this order?
- ‘getting those viruses into’ is a little informal.
Line 245-247. Long confusing sentence. However, and hence, makes it annoying to read. Find a better term than ‘key usefulness’.
- In in repetition
- Define ‘near confluent lysis’. Also, in this section, add a comment on the uncertainty of the actual host of the virus. A good example are the giant viruses mostly isolated in Acanthamoeba. Is this the real host, or is it just susceptible to infection?
- Delete ‘at one time’
- This is ambiguous. You mean to say that viruses do not have a universal marker gene, but the sentence reads as if they don’t have a 16s rRNA gene. The second half of this sentence could be a new sentence. Do you mean environmentAL cellular diversity?
- Delete therefore. What does it really add?
- Simplify this sentence. (Though, nevertheless etc…)
- Does not make sense
- Iruses typo. Also, should be: In addition, viruses…
- Delete ‘and for many labs’
- ‘that is, as due simply’. Simplify this sentence.
- Desorbed virions desortbed. Is there a better way to phrase this?
- Long complicated sentence.
Got as far as line 641.
- For V) I believe these are known as defective interfering particles.
- Visualise is the wrong word here?
- I don’t believe this is accurate. Negative stained capsids which have no nucleic acid can be viewed with uranyl acetate. Where are you getting this information?
- Missing full stop after reference.
- Change ‘that have’ to ‘containing’ or similar
- Due TO defects
- No need to define primer.
732, 733. Change ‘microbe’ to ‘microbial’
- Remove thus
- ‘a producing cell’s genome’. Sounds awkward. Donor cell?
- Comma after ‘currently’
- Replace ‘e.g. given DNase treatment’ with ‘via DNase treatment’
- No need for the word thus
- Unclear what you mean
- Comma after important
- Full stop needed
Reviewer 2 Report
The manuscript represents a comprehensive overview of the major methods used to identify and characterize soil viruses. The three approaches to analyzing soil viruses are presented with their pros and cons. Where possible, the authors also suggest the ways to overcome certain pitfalls of the described methods. I believe that this piece of work is a valuable contribution to the field of soil virology and that it would be especially useful for those who lack own experience, but are interested to start studying soil viruses.
A few comments about the figures:
Fig 1. Please reduce the size of the figure, the text font is now overly big.
Fig 2. Please reconsider the design of this figure (colors and fonts). When printed black-and-white, it is really difficult to read the text in the lower panels, especially in the "Isolation".
Fig 3. legend, line 443 "resulting in" or "resulting from"?
Minor comments about the text:
line 144: "a notion that can tested" -> "a notion that can be tested",
line 181: "virions virocells" -> "virions and virocells",
line 285: extra "in",
line 301: "A few soil virus-soil microbe host systems" -> "A few soil virus-host systems" or "A few virus-soil microbe systems",
line 368: "free or freed DNA", is it redundant? All free DNA was once freed.
line 380: "iruses" -> "Viruses",
line 431: "the more read depth that is available then the more reliable the consensus sequence" -> "the more read depth is available, the more reliable the consensus sequence is" ?
line 562: "Viruses range in size from a few nanometers". I guess the smallest known virus particle is Porcine circovirus, 17 nm in diameter, while "a few" implies something like 2-4 nm.
line 668: "possesses" -> possess" ?
line 745: "encode represent" -> "encode" or "represent",
line 880: full stop is missing after "perspective",
line 1017: extra space before full stop and extra "in which",
line 1041: remove "with the DNA" ?
Reviewer 3 Report
Trubl and colleagues wrote a comprehensive review on soil viral communities and being stuck by coronaviruses at home, it was enjoying reading this manuscript. Though the manuscript provides new information and potentially contributes to the understanding of soil viruses; however, the manuscript needs some improvements that may improve the readership of this work. Overall, authors have done remarkable work in this review and this manuscript may be considered for publication following a revision.
Title: seems too wordy and boring, but it is up-to authors.
Abstract. It will be useful if the authors provide more information about the key findings of this review. In the present form, it is just a wordy description of general stuff”
Ln 32. Are all soil viruses “ infectious agents”? For instance, one ml of seawater contains billion of viruses, can we call them infectious, given the fact many people go to beaches?
Ln 41. Is it not that apply to bacteria as well, I mean “ physical access”
Figure 2. is blurred?
Ln 119. I think you may want to provide a glossary of key viral terms, many soil scientists are not aware of these, e.g. virions
Ln 164. Is the virus-associated horizontal gene transfer tested in the soil system?
Ln 257. Does plating show virus-infected bacterial colonies or viral colonies?
Ln 259. “Direct plating, enrichment, and concentration” you may want to utilize the sub-titles
Ln 423. “shallow” Is it also true for the bacterial rare biosphere as well?
Ln 447. Can you define “vertical sequencing”
Ln 506-509. It is confusing?
Ln 754. Species system may favor the survival of rare microbes by microbial public good products in soil (" Annual Review of Ecology, Evolution, and Systematics 50 (2019): 145-168.).
Overall, it is a well-written manuscript though too wordy, with much less visual presentations (Figs, Tables). Meanwhile, I also think that there is much less known about soil viruses, that is why, authors have provided too much details, which is remarkable and worth appreciating.
